# Enterobacteriaceae as a Key Indicator of Huanglongbing Infection in *Diaphorina citri*

**DOI:** 10.3390/ijms25105136

**Published:** 2024-05-09

**Authors:** Xing-Zhi Duan, Guo-Sen Guo, Ling-Fei Zhou, Le Li, Ze-Min Liu, Cheng Chen, Bin-Hua Wang, Lan Wu

**Affiliations:** School of Life Science, Nanchang University, Nanchang 330022, China

**Keywords:** *Diaphorina citri*, Huanglongbing, Enterobacteriaceae, microbial interactions

## Abstract

Extensive microbial interactions occur within insect hosts. However, the interactions between the Huanglongbing (HLB) pathogen and endosymbiotic bacteria within the Asian citrus psyllid (ACP, *Diaphorina citri* Kuwayama) in wild populations remain elusive. Thus, this study aimed to detect the infection rates of HLB in the ACP across five localities in China, with a widespread prevalence in Ruijin (RJ, 58%), Huidong (HD, 28%), and Lingui (LG, 15%) populations. Next, microbial communities of RJ and LG populations collected from citrus were analyzed via 16S rRNA amplicon sequencing. The results revealed a markedly higher microbial diversity in the RJ population compared to the LG population. Moreover, the PCoA analysis identified significant differences in microbial communities between the two populations. Considering that the inter-population differences of Bray–Curtis dissimilarity in the RJ population exceeded those between populations, separate analyses were performed. Our findings indicated an increased abundance of Enterobacteriaceae in individuals infected with HLB in both populations. Random forest analysis also identified Enterobacteriaceae as a crucial indicator of HLB infection. Furthermore, the phylogenetic analysis suggested a potential regulatory role of ASV4017 in Enterobacteriaceae for ACP, suggesting its possible attractant activity. This research contributes to expanding the understanding of microbial communities associated with HLB infection, holding significant implications for HLB prevention and treatment.

## 1. Introduction

Endosymbiotic bacteria are prevalent in arthropods, exerting a profound influence on the ecology and behavior of their hosts [1,2]. Indeed, they modulate the provision of vital nutrients, regulate reproduction, confer resistance to natural enemies, and drive climate adaptation [3,4,5,6]. Despite the recognized importance of symbionts, the potential interactions influencing their occurrence remain underexplored. Notably, asymmetrical interactions have been observed, such as those between *Spiroplasma* and *Wolbachia* in *Drosophila melanogaster* [7]. In *Curculio sikkimensis*, infections with *Serratia* and *Wolbachia* exhibited a negative correlation [8]. The incidence of such interactions carries significant ecological implications, and elucidating their relationships holds significance for controlling pathogenic bacteria transmitted by vector insects.

The Asian citrus psyllid (ACP), *Diaphorina citri* Kuwayama, a member of the order Hemiptera and family Psyllidae, poses a threat to the Rutaceae family, including citrus, and serves as a vector for the transmission of Huanglongbing (HLB) [9,10]. The pathogen of HLB, known as *Candidatus* Liberibacter asiaticus (CLas), is a highly parasitic Gram-negative bacterium belonging to the alpha subdivision of the Proteobacteria [11]. CLas extensively infects citrus varieties, reducing yields and shortening the lifespan [12]. It is metaphorically described as the “cancer” of citrus plants, presenting significant challenges to the industry. Upon feeding on the phloem of HLB-infected plants, ACP acquires CLas [13]. As an obligate pathogen, CLas is confined to a host-specific environment. By undergoing extensive genome reduction, the CLas genome shrinks to only 1.23 Mb, resulting in the loss of core metabolic pathways [14]. However, various endosymbiotic bacteria coexist with CLas, facilitating the provision of nutritional and metabolic products and concomitantly supporting the survival and reproduction of CLas. Previous studies have investigated fluctuations in the microbial communities of ACP by allowing them to feed on HLB-infected plants in a laboratory [15,16]. However, there exists a substantial difference between laboratory and field environments. In the former, the diversity of the microbial communities of ACP is significantly lower [17], limiting the microbial sources available for identifying interactions with CLas. Therefore, it is essential to conduct surveys on microbial communities in wild populations.

At present, the predominant approach to managing vector insects heavily relies on chemical insecticides, raising concerns about environmental pollution and developing resistance over long-term use [18]. Nevertheless, symbiotic bacteria harbor the potential for effectively managing vector insects [19]. The most crucial step is detecting interactions between symbiotic bacteria and pathogens within vector insects. In the present study, five wild populations of ACP were collected to systematically investigate the role of symbiotic bacteria in the occurrence of HLB. The infection rates of HLB were determined for each population, and those with high infection frequencies were selected for 16S rDNA amplicon sequencing. This study aimed to uncover the interactions and roles of symbiotic bacteria and CLas by examining the microbial communities within HLB-infected and un-infected ACP. The findings hold significant scientific implications and potential practical value for preventing and controlling the transmission of HLB.

## 2. Results

### 2.1. Microbial Community Diversity Associated with Populations and HLB Infection

To identify high HLB infection frequencies across the five populations, 50 adults in each population underwent species-specific PCR detection (Figure 1, Table 1). Following diagnostic PCR, HLB was found to be prevalent in three populations, namely RJ (58%), HD (28%), and LG (15%). Following this, female adults were selected from RJ and LG populations and collected from citrus, with individuals classified as either infected or un-infected with HLB, to explore the impact of HLB on microbial communities.

A total of 9,664,555 high-quality sequences were acquired from 113 samples. After excluding *Candidatus Profftella* sequences and applying sequence rarefaction based on the minimum number of sequences, a comparative analysis of α-diversity indices was performed, including Ace, Chao1, Simpson, and Shannon, between populations with and without HLB infection (Figure 2a–d). Significant differences were observed between populations, although no significant differences were found between infected and un-infected samples within the same population. Additionally, PCoA analysis based on ASVs was employed to assess differences in microbial composition (Figure 2e). While some overlaps were noted between the two populations, PCoA analysis unveiled different microbial community structures. Subsequent statistical analyses, including ANOSIM and Adonis, corroborated these results, demonstrating significant differences in microbial composition between the two populations (ANOSIM, R = 0.305, *p* = 0.001, Adonis, R^2^ = 0.231, *p* < 0.001, Figure 2e,f). Furthermore, the Bray–Curtis dissimilarity within the RJ population exceeded that between groups. In contrast, in the LG population, Bray–Curtis dissimilarity was smaller than that between groups (Figure 2f). These findings collectively demonstrated the significant influence of geographical factors on microbial composition. For a more detailed analysis of the effect of HLB on microbial communities, it is essential to conduct separate analyses for each population.

### 2.2. The Effects of HLB on the Microbial Communities

To detect differences in the microbial community between HLB-infected and uninfected samples in the two populations, a comprehensive analysis of microbial composition was carried out (Figure 3a and Table 2). In addition to the *Candidatus Profftella* sequences, ACP was predominantly populated by the bacterial genus *Wolbachia*, constituting nearly 90% of the overall proportion. On the one hand, *Bacillus*, *Pseudomonas*, *Acinetobacter*, and *Lactobacillus* were more abundant in the RJ population. On the other hand, *Wolbachia* and *uncultured_d_Bacteria* were more abundant in the LG population (Table 2).

Microbial sharing and differential analysis revealed that, compared with the un-infected and infected HLB samples of the LG population, their counterparts in the RJ population shared 144 and 97 genera, acquired 25 and 184 genera, and lost 119 and 135 genera, respectively (Figure 3b). Additionally, when HLB was acquired, HLB un-infected individuals of the RJ and the LG population shared 144 and 103 genera, acquired 119 and 66 genera, and lost 137 and 129 genera (Figure 3b), highlighting significant changes in the microbial communities. Comparing the read percentages of bacterial 16S rRNA genes exposed the fact that the abundance of *Erysipelatoclostridium*, *uncultured_f_Enterobacteriaceae*, *Escherichia-Shigella*, *4-29-1*, *Candidatus Carsonella*, *uncultured_p_Desulfobacterota*, and *Lactobacillus* was significantly higher in the HLB-infected samples of the RJ population. At the same time, the abundance of *SJA-15* was significantly higher in samples without HLB (Figure 3b). Lefse analysis identified enrichments in Enterobacteriaceae and Enterobacterales in the HLB-infected samples (Figure 3). Regarding the LG population, *Candidatus_Solibacter*, *uncultured_o_Enterobacterales*, *Pedosphaeraceae*, *Lawsonella*, *Stenotrophomonas*, and *Ellin6067* were significantly more abundant in non-HLB individuals, whilst the abundance of *uncultured_f_Enterobacteriaceae* was higher in those with HLB (Figure 3c). Importantly, Lefse analysis confirmed enrichments in Enterobacteriaceae and Enterobacterales in the HLB-infected samples of the two populations (Figure 3e,f). Those findings conjointly demonstrated the inconsistent impact of HLB infection on microbial communities, with Enterobacteriaceae consistently elevated in the infected samples.

### 2.3. Microbial Markers Associated with HLB Infection

To elucidate the key genera serving as biomarker taxa in the context of HLB infection, a random forest model was generated to identify the differences between HLB-infected and non-infected samples in the discovery cohort within the RJ population. Through a 10-fold cross-validation with five repeated iterations, the model identified 15 key genera (Figure 4a and Figure 5b). Of note, *uncultured_f__Enterobacteriaceae* emerged as the most important species for discriminating the samples afflicted with HLB (Figure 4b). Thereafter, the probabilities for the discovery cohort were calculated on the 15 genera. In the discovery cohort, there was a significant increase in the probability of the HLB-infected samples from RJ compared to that in the HLB un-infected samples from RJ (*p* < 0.001, Figure 4c), with the probability yielding a remarkable area under the receiver operating characteristic (ROC) curve (AUC) of 1 (*p* < 0.001, Figure 4e). Moreover, to validate the diagnostic efficacy of the classifier model, it was applied to the validation cohort (LG population). The results demonstrated a substantial increase in the probability of the HLB-infected samples from LG compared to that of the HLB un-infected samples from LG (*p* < 0.001, Figure 4d), with the probability achieving an AUC value of 1 (*p* < 0.0001, Figure 4f). Taken together, these results strongly corroborate the diagnostic potential of microbial markers in identifying HLB infected samples, with *uncultured_f__Enterobacteriaceae* emerging as a key biomarker.

### 2.4. Microbial Community Function Related to Populations and HLB Infection

To examine functional alterations within microbial communities with HLB in the two populations, functional predictions were performed using PICRUSt2 analysis. The bacteria identified were enriched in functions associated with signaling and cellular processes, genetic information processing, carbohydrate metabolism, amino acid metabolism, protein family metabolism, and so on (Figure 5a). As anticipated, the abundance of predicted functions significantly varied between the two populations (ANOSIM, R = 0.268, *p* = 0.001, Adonis, R^2^ = 0.194, *p* = 0.001, Figure 5b). Specifically, there were significant differences in terms of the 39 main functions involved in the cellular process, environmental information processing, genetic information processing, and so on (Figure 5c). Analyzing the relationship between the Enterobacteriaceae species and CLas uncovered a significant correlation associated with ASV4017 and ASV1213 (Appendix A). The three primary functions associated with nitrogen fixation, attractant activity, and assistance in insect defense were identified by consulting the literature related to Enterobacteriaceae. Subsequently, gene sequences of relevant species were downloaded from NCBI to conduct a phylogenetic analysis comparing them with ASV4017 and ASV1213. The analysis determined that ASV4017 from ACP clustered together with *Citrobacter* sp. from fruit flies (Figure 5d), its potential attractant activity. However, ASV1213 displayed close evolutionary relationships with several species, and it is challenging to speculate on its specific function (Figure 5d).

## 3. Discussion

To the best of our knowledge, this is the first study to demonstrate that HLB infection in ACP significantly increases the abundance of Enterobacteriaceae in wild populations. Noteworthily, this infection significantly alters the microbial communities of ACP, with Enterobacteriaceae identified as a key biomarker. HLB-infected ACP can pose a significant threat to citrus production, contributing to the transmission of HLB worldwide [10]. The usage of chemical insecticides upon the detection of HLB-infected individuals results in environmental pollution [18]. Hence, enterobacteriaceae could serve as a potential target for interacting with CLas, offering a promising way to prevent HLB.

Enterobacteriaceae, a group of Gram-negative microorganisms, are extensively distributed globally and can thrive in various environments, including the guts of humans, animals, and plants, as well as in soil or water [20,21,22,23,24]. In particular, these bacteria can inhabit the gut of insects, playing crucial roles such as participating in the metabolism and providing nutrients [25,26], exhibiting potential attractive activity [27], and assisting the host in defending against pathogenic microorganisms [28]. However, these studies merely assessed the impact of Enterobacteriaceae on hosts, with no study undertaken into the role of the increased levels of Enterobacteriaceae during the process of vectoring pathogens. Herein, phylogenetic analysis was conducted and it demonstrated that two key species from Enterobacteriaceae, namely ASV4017 and ASV1213, were significantly correlated with species associated with HLB. Interestingly, ASV4017 clustered with Citrobacter sp. strains, highlighting its potential attractant activity. Enterobacteriaceae can synthesize unique chemical substances that possess attractant activity. Earlier research reported that clustered Enterobacteriaceae isolated from the gut of the apple maggot fly generated higher levels of indole, 3-methyl-1-butanol, and 2,5-dimethylhydrazine. At the same time, strains isolated from the Mexican fruit fly produced 3-hydroxybutanone, but only indole demonstrated attractive activity in adult flies, whereas other chemical substances inhibited the attractive activity of indole [29]. Another study identified several Enterobacteriaceae bacteria isolated from the reproductive glands of female oriental fruit flies with attractive activity to female adults, including *Klebsiella oxytoca*, *Klebsiella pneumoniae*, and *Raoultella terrigena*. In field cage experiments, *Klebsiella oxytoca* attracted the highest number of female adults [30]. HLB-infected ACP can attract uninfected individuals to aggregate on plants with HLB [18]. Nonetheless, the role of Enterobacteriaceae in driving this attraction warrants further investigation.

Herein, despite considerable differences in microbial communities between the two populations, Enterobacteriaceae were significantly enriched in HLB-infected ACP, consistent with findings from a previous laboratory study [15]. Enterobacteriaceae in relation to HLB infection are not fully understood. CLas can improve the growth environment for Enterobacteriaceae. Such potential mechanisms can include the production of antimicrobial compounds, altering the host’s immune response, or affecting the host’s nutritional acquisition. Further research will be necessary to confirm these possibilities and to understand their impact on the HLB progression. In addition to Enterobacteriaceae, our field population studies revealed that each population harbored some particular symbiotic bacteria interacting with HLB, indicating that differences in symbiotic bacteria could result from different environments. Significant differences were observed in the microbial composition of field-collected ACP compared to the populations studied in the two previous studies, wherein changes in the microbial communities were investigated by allowing them to feed on HLB-infected plants in the laboratory setting [15,16]. Similar findings were previously documented in other insect species, such as the *Drosophila* [31] and the *Triatoma infestans* [32]. According to another study, ACPs in the laboratory exhibited more robust connectivity and intricate network structure within their microbial communities than in natural populations [17]. Taken together, our observations suggest that differences in microbial communities could be linked to environmental variability.

Our results indicated that, besides *Candidatus Profftella*, *Wolbachia* is the most predominant genus in wild ACP. *Wolbachia* is currently regarded as the most widely distributed secondary endosymbiont [33], known to regulate host reproduction through four mechanisms, namely cytoplasmic incompatibility (CI), parthenogenesis, male-killing, and feminization [34]. Additionally, *Wolbachia* can influence various host functions, encompassing growth, development, lifespan, and nutrient acquisition [35,36]. In ACP populations, the infection rates of *Wolbachia* exceeded 90% [37,38]. Furthermore, the abundance of *Wolbachia* was positively correlated with CLas [39]. However, our findings did not identify any interactive effects between *Wolbachia* and CLas. This could be ascribed to the influence of environments on the abundance of *Wolbachia*. During our sample collection in the summer, the high temperatures may have lowered *Wolbachia* titers [40], influencing their interactions with CLas. *Wolbachia* also regulates host insect gene expression, thereby creating an optimal internal environment conducive to *Wolbachia* proliferation. While the reproductive implications of *Wolbachia* in ACP remain unexplored, investigating the reproductive patterns of *Wolbachia*-harboring ACP could unravel a mechanism contributing to the heightened *Wolbachia* abundance in the presence of Las [38]. Beyond its influence on host gene expression, *Wolbachia* has a protective role against a broad spectrum of pathogens [41]. Intriguingly, this bacterium exhibits a probiotic effect concerning HLB. Moreover, prior investigations portrayed *Wolbachia*’s potential to modulate host immunity, with implications for vector-borne pathogens [42]. This dual role of *Wolbachia*, both as a protector and potential enhancer of certain infections, underlines its intricate interplay within host systems.

Herein, *Candidatus Profftella* stood out as the most abundant genus within ACP, which inhabited the central syncytial cytoplasm of the bacteriome, functioning as a defensive symbiont [43]. In addition, *Candidatus Profftella* was found to be co-exclusive with CLas. Genomic and phylogenetic analyses unraveled a fascinating interplay, showing a horizontally acquired gene in the CLas lineage originating from *Candidatus Profftella* and highlighting complicated ecological and evolutionary connections between these units [44,45]. Previous research evinced that the abundance of *Candidatus Profftella* was higher in ACP laboratory populations compared to that in field-collected populations, demonstrating the varying responses to different environments [17]. Due to its remarkably high abundance, its sequences were excluded from our analysis. Therefore, a more detailed study is necessary to elucidate its specific roles.

In the current study, ACP was sampled from five populations in China, revealing three populations with relatively high rates of HLB infection. Given that the RJ and LG populations were collected from citrus, we focused on these two populations to investigate the impact of HLB on the microbial communities of ACP. In total, 16S rDNA amplicon sequencing determined that the RJ population exhibited significantly higher microbial diversity than the LG population. For the RJ population, differences in the Bray–Curtis dissimilarity within populations exceeded those between populations. Consequently, separate analyses were conducted for each population. Our results illustrated that Enterobacteriaceae was particularly more abundant in HLB-infected individuals in both populations. Likewise, random forest analysis displayed that Enterobacteriaceae played a decisive role in HLB infection. Further predictions of microbial community functions highlighted significant differences between populations. Lastly, phylogenetic analysis suggested that ASV4017 from Enterobacteriaceae may have potential attractant activity. This study investigated microbial communities that interact with CLas, providing a theoretical reference for the prevention and treatment of HLB.

## 4. Materials and Methods

### 4.1. Sample Collection

Between August and October 2022, female ACP adults were collected from five distinct citrus cultivation sites in China (Figure 1 and Table 1). Each ACP sample was collected at intervals exceeding 1 m to avoid sampling closely related individuals. All adults were preserved in undiluted ethanol and maintained at −20 °C until DNA extraction.

### 4.2. HLB Infection Detection

All ACP adults underwent a triple rinse with sterile water before DNA extraction. The DNA of each individual was extracted and purified using the QIAGEN DNeasy kit (QIAGEN, Hilden, Germany) following the manufacturer’s protocol. The quality and concentration of the DNA samples were assessed using a NanoDrop 2000 spectrophotometer (Thermo Fisher Scientific, Wilmington, NC, USA). Qualified DNA samples were subsequently stored at −20 °C for the subsequent experiments. To assess the infection rates of HLB, 50 adults were randomly selected from each population for PCR. The nested PCR consisted of two rounds, with primer sequences and fragment lengths detailed in Appendix A [38]. For the first round, the extracted DNA served as a template, with a reaction volume of 20 μL and consisting of 1 μL of DNA (100 ng/mL), 10 μL of 2× Hieff PCR Master Mix (Yeasen, Shanghai, China), 0.8 μL of 27F/1492R primers (10 μmol each), and 8.2 μL of sterile water. The amplification conditions were as follows: initial denaturation at 94 °C for 5 min, followed by 20 cycles of denaturation at 94 °C for 30 s, annealing at 55 °C for 30 s, extension at 72 °C for 1 min for 30 s, and final extension at 72 °C for 10 min. For the second round, using a 1000-fold dilution of the first-round product as a template, the reaction volume was 20 μL and comprised 1 μL of DNA, 10 μL of 2× Hieff PCR Master Mix, 0.8 μL of OI1/OI2 primers (10 μmol each), and 8.2 μL of sterile water. The amplification conditions were as follows: initial denaturation at 94 °C for 5 min, followed by 35 cycles of denaturation at 94 °C for 30 s, annealing at 57 °C for 30 s, extension at 72 °C for 1 min 20 s, and final extension at 72 °C for 10 min. Positive and blank controls were included in the PCR process, with the former consisting of DNA samples that had been previously amplified and the latter composed of sterile water. The samples of HLB infection were identified by analyzing the PCR products on a 1% agarose gel. Based on the infection rates (Figure 1), the RJ and LG populations were selected to explore the effect of HLB on microbial communities. 

### 4.3. 16S rDNA Amplicon Sequencing

In total, 33 infected and 27 non-infected HLB samples were selected within the RJ population. Similarly, 17 infected and 36 non-infected HLB samples were selected within the LG population for subsequent 16S rDNA amplicon sequencing. PCR specifically amplified the V3-V4 region of the 16S rDNA sequence, with primers listed in Appendix A [17]. The reaction volume for PCR was 50 μL, comprising 2 μL of DNA (50 ng/mL), 25 μL of 5 × FastPfu buffer (Sangon, Shanghai, China), 3 μL of 338F/806R primers (10 μmol each), and 20 μL of sterile water. The amplification conditions were as follows: initial denaturation at 95 °C for 3 min, followed by 27 cycles of denaturation at 95 °C for 30 s, annealing at 55 °C for 45 s, extension at 72 °C for 30 s, and final extension at 72 °C for 10 min. Both positive and blank controls were included in the process. To assess the efficiency of amplification, 5 μL of the products were loaded onto a 1.5% agarose gel, with a 2000 Marker serving as a reference during gel electrophoresis. Afterward, the PCR products were purified using the Cycle Pure Kit (OMEG, Norcross, GA, USA). Following purification, the products were dispatched to the Personalbio company (Shanghai, China) for 16S rDNA amplicon sequencing on the Illumina Miseq 2500 platform with a paired-end configuration (2 × 250 bp). The raw reads of 16S rRNA sequencing have been deposited in the NCBI Sequence Read Archive (SRA) database.

### 4.4. Bioinformatics Analysis

The raw sequencing data acquired from the Illumina Miseq 2500 platform were processed and analyzed using the QIIME2 platform [46]. Trimmomatic-0.36 was employed to exclude ambiguous and low-quality sequences from the raw data [47]. Sequence merging was accomplished using FLASH 1.2.11 [48]. Primer and chimeric sequences were identified and discarded using cutadapt 1.8 and Usearch 9.2 [49]. Amplicon sequence variants (ASVs) were clustered using Usearch 9.2 [50]. Taxonomic annotation of ASVs was conducted using the SILVA 138 database [51]. Attributed to the high proportion of *Candidatus Profftella* sequences, this genus was excluded from the downstream analysis. Subsequently, the sequencing depth of all samples was normalized to the minimum number of sequences within samples employing the R “vegan” package. The α-diversity and the Bray–Curtis dissimilarity were calculated using the R “vegan” package [52]. The significance of differences in the α-diversity index was assessed by Duncan’s new multiple tests. The relative abundance (%) for each genus within each group was determined by calculating the number of reads assigned to that genus relative to the total number of reads within each group. Principal Coordinate Analysis (PCoA) was utilized to visualize the differences between samples [53]. Differential abundance of microbial features was evaluated using Linear Discriminant Analysis Effect Size (LEfSe) (http://galaxy.biobakery.org/ (accessed on 1 April 2024)) [54] and STAMP with Mann–Whitney U tests [55,56]. Statistical differences in microbial communities between different groups were examined using ADONIS and ANOSIM in the R “vegan” package [57]. The predictive model was constructed using random forest analysis and implemented using the R “randomForest” package with default parameters [58]. The training dataset for the model was selected from the RJ population, whilst the dataset from the LG population was used for model validation. The cutoff point for the model was determined by the minimum point on the cross-validation error curve, obtained through five iterations of 10-fold cross-validation experiments. The ROC curve was constructed to evaluate the performance of the model, and the AUC value was used to represent the ROC effect by the R “pROC” package [38]. The Phylogenetic Investigation of Communities by Reconstruction of Unobserved Species2 (PICRUSt2) was applied to investigate microbial potential function [59]. The phylogenetic analysis of Enterobacteriaceae was constructed based on the 453 bp length of the 16S rRNA gene using Mrbayes 3.2.6 [60].

## Figures and Tables

**Figure 1 ijms-25-05136-f001:**
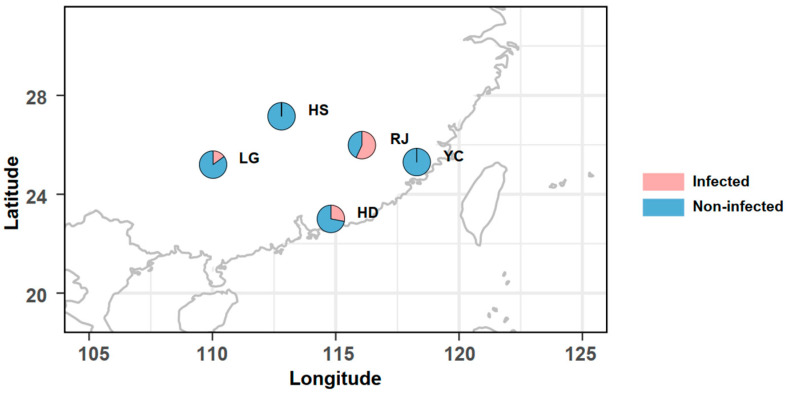
Sampling localities and infection frequencies of HLB in the five ACP populations. The detailed information is provided in Appendix A.

**Figure 2 ijms-25-05136-f002:**
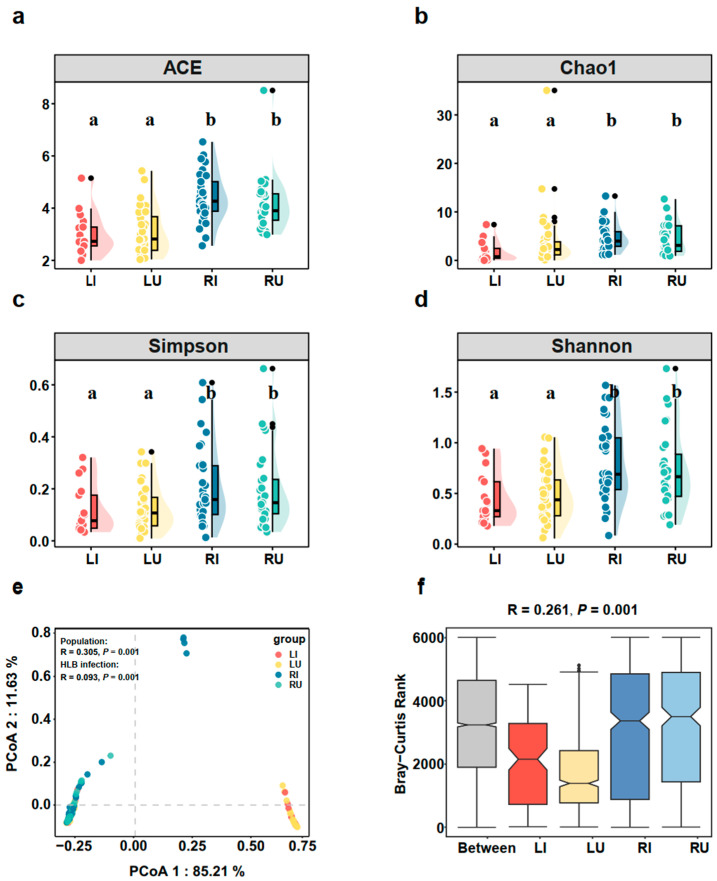
Comparison of the microbial communities between the infected and un-infected samples of the two populations. (**a**) ACE index, (**b**) Chao1 index, (**c**) Simpson index, (**d**) Shannon index, (**e**) PCoA analysis, and (**f**) the ANOSIM analysis. The combined charts featuring a scatter plot and a bar chart. The scatter plots illustrated diversity distribution within each group, differentiated by color. The bar chart complemented this with a summary of average diversity per group, represented by bar height, and variability, shown by error bars extending from the first to the third quartile. Significance was assessed using Duncan’s new multiple tests, with distinct letters representing different levels of significance. The PCoA generated by Bray–Curtis dissimilarity was based on ASVs. Regarding ANOSIM analysis, the x-axis denotes within- or between-population comparisons, whilst the scale on the y-axis represents the magnitude of the distance value. RI, HLB infected samples from RJ; RU, HLB un-infected samples from RJ; LI, HLB infected samples from LG; LU, HLB un-infected samples from LG.

**Figure 3 ijms-25-05136-f003:**
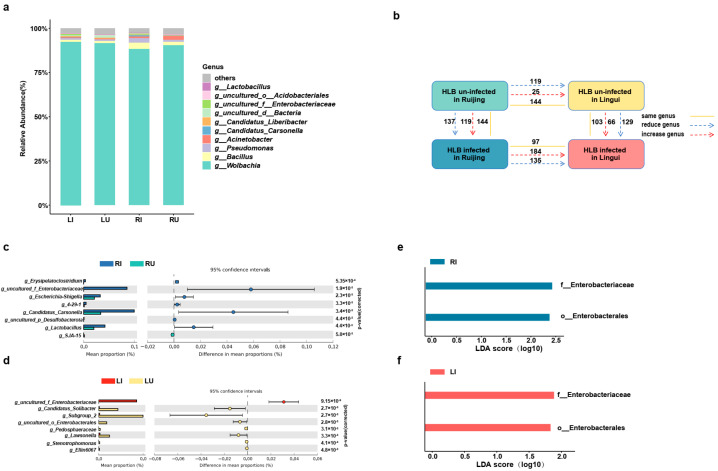
Differences in the microbial community between the HLB-infected and un-infected samples of the two populations. (**a**) The relative abundance of bacterial 16S rRNA genes, highlighting the top 10 genera. (**b**) Shared, reduced, and increased genera in the four groups. The direction of the arrows and the numbers next to them represent changes in the number of genera. Yellow solid lines, shared genera; blue dashed lines, reduced genera; red dashed lines, increased genera. (**c**,**d**) The read percentages of bacterial 16S rRNA genes were presented as relative abundance (%) for the RJ and LG populations. Statistical significance was assessed using the Mann–Whitney U test, with error bars indicating 95% confidence intervals. (**e**,**f**) Significant differences revealed by Lefse analysis in microbial communities (LDA > 1.5) between the two populations. RI, HLB-infected samples from RJ; RU, HLB un-infected samples from RJ; LI, HLB-infected samples from LG; LU, HLB un-infected samples from LG.

**Figure 4 ijms-25-05136-f004:**
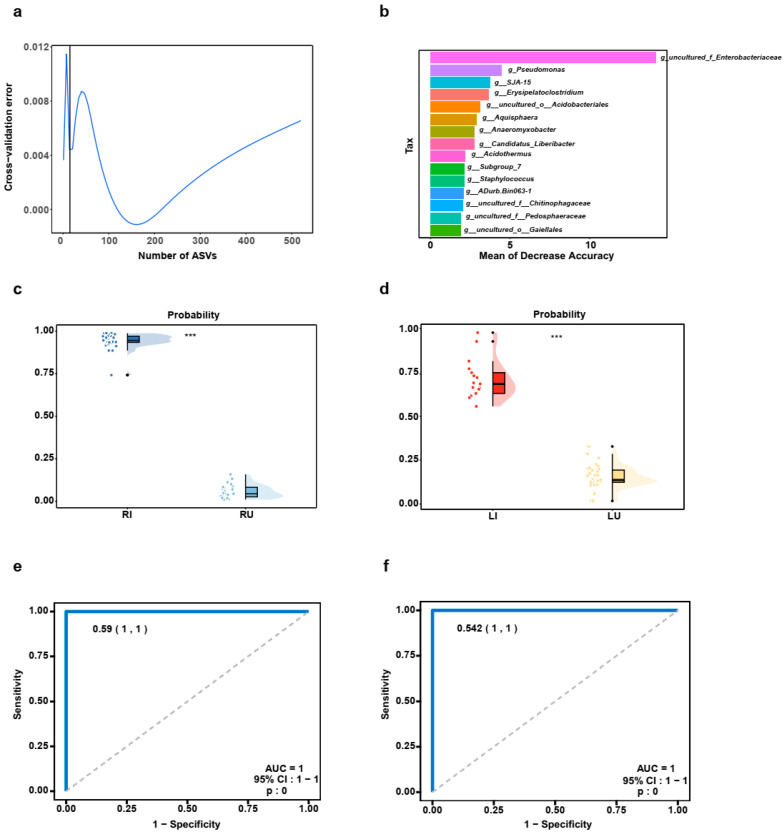
Microbial markers associated with HLB infection in the two populations. (**a**) Cross-validation error with the number of genera. (**b**) Biomarker taxa based on importance in accurately predicting the effects of HLB on the microbial communities. (**c**) The probability in the RI group (*n* = 33) versus the RU group (*n* = 27) in the discovery cohort. (**d**) The probability in the LI group (*n* = 17) versus the LU group (*n* = 36) in the validation cohort. (**e**) AUC value based on the probability in the discovery cohort. (**f**) AUC value based on the probability in the validation cohort. The combined charts featuring a scatter plot and a bar chart. The scatter plots illustrated probabilities within each group, differentiated by color. The bar chart complemented this with a summary of average diversity per group, represented by bar height, and variability, shown by error bars extending from the first to the third quartile. ***, *p* < 0.001.; RI, HLB-infected samples from RJ; RU, HLB un-infected samples from RJ; LI, HLB-infected samples from LG; LU, HLB un-infected samples from LG; CI, confidence interval; AUC, area under the curve.

**Figure 5 ijms-25-05136-f005:**
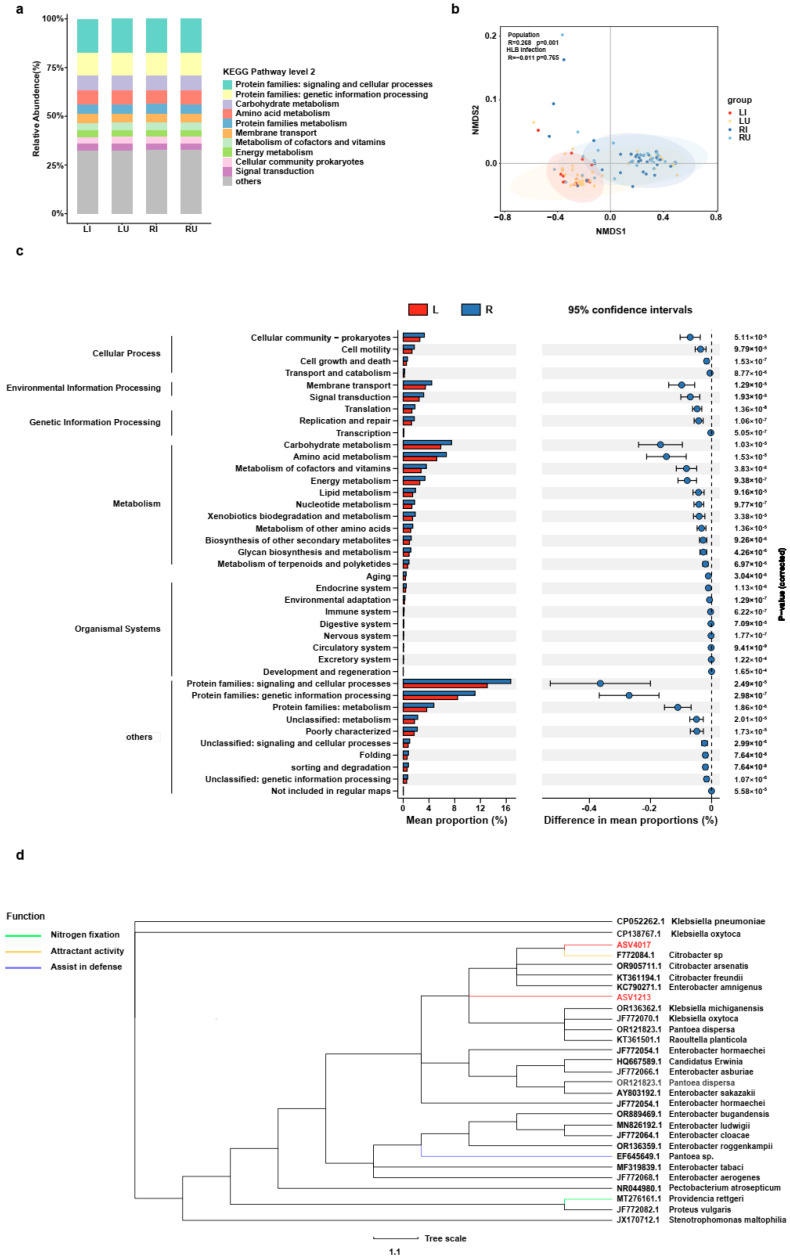
Function prediction of microbial communities between HLB-infected and un-infected samples of the two populations and phylogenetic analysis involving multiple species within Enterobacteriaceae. (**a**) The relative abundance of predicted functions based on KEGG pathway level 2, highlighting the top 10 functions. (**b**) Non-metric Multidimensional Scaling (NMDS). NMDS was generated by Bray–Curtis dissimilarity based on ASVs. (**c**) The read percentages of predicted functions are presented as relative abundance (%) for the RJ and LG populations. Statistical significance was assessed using the Mann–Whitney U test, with error bars indicating 95% confidence intervals. (**d**) Evolutionary analysis of the Enterobacteriaceae strain from ACP. The phylogenetic tree was constructed based on the 453 bp length of the 16S rRNA gene using Mrbayes 3.2.6. The red lines represented ASV4017 and ASV1213 from Enterobacteriaceae. The other Enterobacteriaceae 16S sequences were from insects, sourced from the NCBI database. The scale bar represents percentages of bootstrap replications and sequence divergence calculated from 1000 replicate trees. NCBI sequence accession numbers and the names of the corresponding species are presented. The functions of different species were identified by consulting the relevant literature. *Klebsiella pneumoniae* (CP052262.1) and *Klebsiella oxytoca* (CP138767.1) were used as three outgroups. RI, HLB-infected samples from RJ; RU, HLB un-infected samples from RJ; LI, HLB-infected samples from LG; LU, HLB un-infected samples from LG.

**Table 1 ijms-25-05136-t001:** Summary of collection details, including the population code (ID), province, city, latitude, longitude, temperature, collection date, and the CLas infection frequency.

ID	Province	City	County	Host	Latitude	Longitude	Temperature	Date	Frequency
HS	Hunan	Hengyang	Hengshan	Orange	27.1 °C	112.8 °C	32 °C	2 October 2022	0
RJ	Jiangxi	Ganzhou	Ruijin	Orange	25.9 °C	116.1 °C	29 °C	3 August 2022	57%
YC	Fujian	Quanzhou	Yongchun	wampee	25.3 °C	118.3 °C	29 °C	4 August 2022	0
LG	Guangxi	Guilin	Lingui	Orange	25.1 °C	110.0 °C	33 °C	3 October 2022	15%
HD	Guangdong	Huidong	Huidong	Wampee, kamuning	23.0 °C	114.8 °C	32 °C	6 August 2022	28%

**Table 2 ijms-25-05136-t002:** Relative abundance (%) of bacterial 16S rRNA genes at the genus level observed for R, RU, L, LU group, and all groups. RI, HLB infected samples from RJ; RU, HLB un-infected samples from RJ; LI, HLB infected samples from LG; LU, HLB un-infected samples from LG.

Genus Classes	LI (%)	LU (%)	RI (%)	RU (%)	All Groups (%)
*Wolbachia*	92.15	91.51	88.26	90.34	89.86
*Bacillus*	1.32	1.26	3.52	1.85	2.38
*Pseudomonas*	0.81	0.85	2.61	1.27	1.7
*Acinetobacter*	0.47	0.34	0.91	1.96	1.09
*Candidatus_Carsonella*	0.27	0.2	0.76	0.25	0.46
*Candidatus_Liberibacter*	0.52	0.42	0.2	0.54	0.37
*uncultured_d_Bacteria*	0.51	1.37	0.02	0.06	0.34
*uncultured_f_ Enterobacteriaceae*	0.57	0	0.63	0	0.3
*uncultured_o_ Acidobacteriales*	0.28	0.31	0.18	0.3	0.25
*Lactobacillus*	0.1	0.1	0.32	0.15	0.2
Others	2.94	3.58	2.54	3.23	2.99

## Data Availability

The data presented and that support the findings of this study are openly available on Zenodo at https://zenodo.org/uploads/10863206 (accessed on 1 April 2024).

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
