# Peer review of "Enterobacteriaceae as a Key Indicator of Huanglongbing Infection in Diaphorina citri"

_ijms, 2024, doi:10.3390/ijms25105136_

Round 1

Reviewer 1 Report

Comments and Suggestions for Authors

The study is devoted to the urgent problem of environmental factors affecting the causative agent of vector-borne disease of citrus fruits -- Huanglongbing (HLB).

The authors found a difference in the microbiomes of intact insects and insect carriers of infectious agents, conducted statistical processing, the results are undoubtedly very interesting.

However, it seems to me that we should still be careful in the interpretation.

The samples contain unidentified taxa marked with сhipers (4-29-1 or Subgroup 2)... What is hidden behind them, do they have an important influence?

The authors rightly noted that geographical location has a greater impact on the composition of communities than the presence of HLB-agent in them. In this case, it is necessary to evaluate the microbiome of the environment more. Insects come into contact with plants in different locations, individual microbiocenoses on the surface of plants - they are in inseparable mutual influence

Line 41

Please clarify the taxonomic position CLas according to https://lpsn.dsmz.de/

Table 3

The group designations in the title and in the table are different --- « ….for R, RU, L, LU group».. and then in table “LI (%)  LU (%)  RI (%)  RU (%)”….

Нow was the relative abundance calculated and what is the “all samples” column?

Lines 350-351

Why were different numbers of individuals chosen for the research?

If initially 50 individuals from each location were selected for research, then why did Ri, RU, LI, LU-sets of individuals were unequal?

Fig.5d

The interpretation of this tree from the standpoint of the functions of Enterobacteriaceae seems to me very speculative: «The functions of different species were identified by consulting the relevant literature» -- representatives of the family are found in animals and plants, everywhere and on dead organic matter, and functions can be diverse

 Fig. S1

Figure captions are missing

Reviewer 2 Report

Comments and Suggestions for Authors

The manuscript is well written and described. This work is of great significance in the field of understanding and mitigating HLB disease.

There are some minor improvements needed to be made for the manuscript to be clear and more scientific.

Figures need to be improved. Fig 3-5 is hardly understandable.

The importance of the enrichment of  Enterobacteriaceae in HLb-infected population should be mentioned in the results section. 

The discussion needs improvement in terms of scientific explanations of the microbial correlation to HLB pathogen and infection.
